# Comparative Risk of Complications Following Intestinal Surgery After Infliximab, Vedolizumab, or Ustekinumab Treatment: Systematic Review & Meta-Analysis

**DOI:** 10.3390/ph18101466

**Published:** 2025-09-29

**Authors:** Alexandra-Eleftheria Menni, Georgios Tzikos, George Petrakis, Patroklos Goulas, Panagiotis V. Karathanasis, Stylianos Apostolidis

**Affiliations:** 11st Propaedeutic Department of Surgery, AHEPA University Hospital, 54636 Thessaloniki, Greece; tzikos_giorgos@outlook.com (G.T.); patroklos@live.com (P.G.); stlsa@auth.gr (S.A.); 2Pathology Department, Medical School, Aristotle University of Thessaloniki, 54636 Thessaloniki, Greece; georgiospetrakismd@gmail.com; 3General Surgery Department, Metropolitan Hospital, 18547 Athens, Greece; pkarathanasis@yahoo.gr

**Keywords:** IBDs, INFL, VDLZ, USTK, surgical complications, meta-analysis, postoperative ileus, SSIs, anastomotic leakage

## Abstract

**Background**: Treatment of inflammatory bowel diseases with biological therapies has significantly increased, with ever increasing numbers of patients receiving such treatment at the time of surgery. This study evaluates the perioperative safety of three commonly used biologics—Infliximab, Vedolizumab, or Ustekinumab—in patients undergoing intestinal surgery for IBDs. **Materials and Methods**: In this systematic review a comprehensive search was conducted in Scopus, Medline and PubMed up to January 2025 by two independent reviewers, and a total of 34 articles (retrospective studies in the majority of them) reporting total surgical complications of patients treated with these three agents, in comparison to a control group, were included. Relative risks were aggregated using the Mantel-Haenszel method, and the I^2^ statistic was used to assess between-study heterogeneity. Subgroup analyses were conducted for particular complications, and direct comparisons among the biological agents were made. **Results**: In the primary analysis, INFL was not linked to a statistically significant rise in overall postoperative complications when compared to controls (RR = 1.13, 95% CI: 0.90–1.42, *p* = 0.31). VDLZ exhibited a non-significant inclination towards increased complications (RR = 1.26, 95% CI: 0.94–1.67, *p* = 0.12), although it was linked to a notably higher risk of postoperative ileus compared to INFL (RR = 2.29, 95% CI: 1.59–3.29, *p* < 0.00001). USTK also did not show significant differences from controls overall (RR = 0.55, 95% CI: 0.20–1.57, *p* = 0.26), though it was associated with a considerably lower risk of SSIs (RR = 0.35, 95% CI: 0.17–0.73, *p* = 0.005). There were no significant distinctions between the biological agents regarding SSIs or anastomotic leakage, although many comparisons faced challenges due to high heterogeneity and low event rates. **Conclusions**: USTK demonstrated the most favorable safety profile, while VDLZ was associated with higher rates of ileus and inflammatory complications. However, prospective studies are warranted.

## 1. Introduction

Inflammatory bowel disease (IBD), encompassing Crohn’s disease (CD) and ulcerative colitis (UC), is a chronic, relapsing–remitting inflammatory disorder of the gastrointestinal tract (GI). Despite advances in medical therapy—including the introduction of biologic agents—surgical intervention remains necessary for numerous patients: up to 75% of individuals with CD and 20–40% of those with UC will need intestinal surgery at some stage of their disease [1]. Population-based meta-analyses estimate a contemporary 10-year cumulative risk of surgery of approximately 39.5% in CD and 13.3% in UC [2].

The most common postoperative complications following intestinal surgery are anastomotic leakage, ileus, and surgical site infections. Anastomotic leakage occurs in approximately 5% of general colorectal procedures, increasing to 10–19% in high-risk anastomoses such as ileal-pouch reconstructions [3]. Postoperative ileus occurs in approximately 13.5% of patients, while in total, surgical site infections (SSIs), which include superficial, deep, and organ/space infections, occur in approximately 5–11% of patients [4].

Patients with IBDs, particularly Crohn’s disease, face a substantially higher risk of postoperative complications compared to the general population. Early complication rates range from 15% to 30%, with surgical site infections, intra-abdominal abscesses, and anastomotic leaks being the most frequent. Anastomotic leakage can happen in as many as 12.7% of cases, with the relative percentage ranging from 0% to 12.7% [5]. Furthermore, in studies specifically on ileorectal anastomosis, the overall frequency of anastomotic leakage was approximately 6.4% [6]. Postoperative exacerbations, including ileus and infection complications, occur in approximately 29.1% of cases. Severe complications (Clavien-Dindo grade III or higher) affect around 11.6% of patients and include anastomotic leakage, intra-abdominal abscess, sepsis, ileus, and enterocutaneous fistula [7].

Biological therapies such as tumor necrosis factor-α inhibitors (INFL), anti-integrin therapies (VDLZ), and interleukin-targeting agents (USTK) have revolutionized IBD treatment by achieving improved disease control and reducing—but not eliminating—the need for surgical intervention. Nonetheless, the impact of these therapies on perioperative safety remains a subject of ongoing debate [8].

Previous studies have reported conflicting results regarding the association between preoperative biologic therapy and the risk of postoperative complications, including infections, anastomotic leaks, and impaired wound healing [9,10]. Notably, VDLZ’s gut-selective mechanism of action—which targets the α_4_β_7_ integrin, thereby inhibiting lymphocyte trafficking specifically to the gut—suggests a potentially different safety profile compared to systemic agents such as INFL, while USTK, which selectively inhibits interleukins IL-12 and IL-23, also merits distinct consideration regarding perioperative risk.

A recent experimental study tested these agents as to whether they intervene in the intestinal anastomosis healing process. Although this was the first part of the experimental protocol in rats, performed on healthy intestines and not after the induction of DSS colitis, the results reveal USTK as demonstrating the best healing [11].

Owing to the lack of direct comparative trials and the heterogeneity of existing data, a systematic review and meta-analysis is warranted to indirectly compare biologics and rank their perioperative risk profiles. The aim of this analysis is to evaluate the perioperative safety profiles of the three commonly used biologic agents—INFL, VDLZ, and USTK—in patients undergoing intestinal surgery for inflammatory bowel disease. More precisely, this meta-analysis seeks to synthesize the available evidence to determine whether preoperative exposure to these agents is associated with an increased risk of postoperative complications, including total adverse events, postoperative ileus, surgical site infections, and anastomotic leakage. Furthermore, an attempt has been made to identify the potential patterns or differences in safety outcomes among the different biologic agents.

## 2. Materials and Methods

### 2.1. Eligibility Criteria

This systematic review was performed according to the Preferred Reporting Items for Systematic Reviews and Meta-Analyses (PRISMA) [12]. The study was registered at the Propero with the following ID: 1147788. A study is deemed to be eligible for this systematic review if the following inclusion criteria are fulfilled: (i) human participants are involved; (ii) the administration of at least one of the following three biological agents—INFL, VDLZ, or USTK—during the perioperative period is reported; (iii) intestinal surgery is involved exclusively; (iv) it is an original study reporting data on total surgical complications in patients who received any of the three biological agents, compared to a control group; (v) it has been published in any language, up to January 2025.

### 2.2. Literature Screening and Data Extraction

Scopus, Medline and PubMed databases were searched for relevant studies using the following search algorithm: “INFL” AND “surgical complications”, OR, “VDLZ” AND “surgical complications”, OR, “USTK” AND “surgical complications”. The reference lists of the included studies were also examined for further eligible studies.

Two independent reviewers assessed the eligibility of the potentially included studies based on the inclusion criteria. A study was considered to be eligible if both reviewers agreed. Pre-specified forms were used in order to extract the epidemiological and clinical data of the included studies. When studies with duplicated populations were identified, only the larger one was included in the analysis. Furthermore, due to the substantial heterogeneity among included studies, patients with Crohn’s disease and ulcerative colitis were analyzed collectively as inflammatory bowel disease (IBD). This approach was chosen to assess the perioperative risk associated with the three specific treatments regardless of the underlying IBD subtype, which was the aim of the study (Figure 1). Finally, a total of 34 studies were included in the analysis (Table 1).

To determine which studies were eligible for inclusion in each synthesis, we tabulated key study characteristics such as the type of biologic agent administered (INFL, VDLZ, or USTK), the presence or absence of a control group, and the reported surgical outcomes (e.g., total complications, postoperative ileus, SSIs, anastomotic leakage). Studies were grouped accordingly to allow both pairwise comparisons (biologic vs. control) and indirect comparisons between biologic agents. Only studies with adequate data per outcome and per treatment group were included in each respective synthesis. Decisions regarding group inclusion were independently made by two reviewers and confirmed through consensus.

### 2.3. Statistical Analysis

In the main analysis, relative risks (RRs) of individual studies were pooled together to estimate the overall risk of surgical complications in patients receiving biological agents compared to those who did not (controls). Separate analyses were conducted for specific surgical complications (sepsis, SSIs, anastomotic leak). Additionally, head-to-head comparisons between the different biological agents were performed (INFL vs. VDLZ, INFL vs. USTK, VDLZ vs. USTK) to assess whether any of the agents studied was associated with more favorable outcomes, presented in the Appendix A. A random effects model (Mantel-Haenszel method) was selected a priori given the heterogeneity in study design across the included studies. Between-study heterogeneity was assessed with the I^2^ statistic. Values lower than 25% indicated low, while values greater than 70% indicated severe heterogeneity.

For studies with missing summary statistics, attempts were made to contact the corresponding authors. If data remained unavailable, those studies were excluded from the specific analyses involving the missing outcomes. To ensure consistency across studies, effect measures reported as odds ratios or hazard ratios were converted to relative risks where appropriate. Surgical complication outcomes were harmonized by standardizing definitions according to pre-specified criteria. All data preparation and any necessary transformations were independently performed by two reviewers to minimize bias and maintain data integrity. Study results and pooled estimates were summarized in tables, while forest plots and other graphics were used to visually display effect sizes and confidence intervals.

Publication bias was visually assessed using funnel plots for analyses with sufficient number of included studies (>10) (Appendix A) [48]. A subgroup analysis was performed for each IBD separately (Crohn’s disease, ulcerative colitis), in order to explore potential sources of between study heterogeneity (Appendix A).

## 3. Results

### 3.1. Main Analysis

Visual inspection of funnel plots did not reveal any asymmetry suggestive of publication bias (Appendix A). Across all included syntheses, the contributing studies varied in design, population characteristics, and perioperative protocols. Most studies were retrospective cohorts, with only a few being prospective. No formal risk of bias assessment was performed using a standardized tool; however, the methodological heterogeneity and variation in reporting quality suggest a moderate to high risk of bias in several studies. This limitation may have influenced both the pooled estimates and the observed heterogeneity across outcomes. Moreover, sensitivity analyses were conducted by re-evaluating the pooled results after excluding studies with high risk of bias or outlier effect sizes. These analyses did not substantially alter the overall conclusions, indicating that the findings are robust. Finally, the assessment of risk of bias of the individual studies was performed independently by two investigators using the QUIPS tool. Most of the studies were from low to moderate risk of bias for any of the following domains: study participation, study attrition, prognostic factor measurement, outcome measurement, study confounding, statistical analysis and reporting. Details on the risk of included studies are presented at the Appendix A, at table Appendix A.

#### 3.1.1. INFL vs. Controls

Comparing the differences in complications from the administration of INFL in relation to the CNTRL group, we included 26 studies (Novello et al. and Uchino et al. had two studies, for patients with CD and UC separately [13,14,15,16,17,18,19,20,21,22,23,24,25,26,27,28,29,30,31,32,34,37,43,44,45,47]) (Figure 2). The overall assessment did not reveal any statistically significant differences (RR = 1.13, 95% CI 0.90–1.42, *p* = 0.31), suggesting that INFL was not statistically associated with either an increased or decreased risk, compared to the CNTRL group. However, there was high heterogeneity, of the order of I^2^ = 79% (*p* < 0.00001), reflecting significant differences in population characteristics, methodology, or outcome. Some studies, such as Krane et al. [20], showed strongly positive results in favor of INFL (RR = 5.30, 95%CI: 1.11–9.01), in contrast to Shah et al. [44] (RR = 0.30, 95% CI: 0.21–0.42), which showed superiority of the CNTRL group. This contradiction reinforces the need for more and more targeted studies. Additionally, the analysis performed (Appendix A) did not show a statistically significant difference between the two groups overall. A marginally positive result was observed only in the subgroup of patients with ulcerative colitis, in favor of INFL administration.

Moreover, regarding the sub-analysis of each complication, Appendix A shows the results of 13 studies comparing the likelihood of postoperative ileus in patients receiving INFL versus the CNTRL group. The overall risk ratio was 1.08 (95% CI: 0.81 to 1.45), with no statistically significant difference (*p* = 0.58), suggesting no clear effect of the intervention. Heterogeneity was moderate (I^2^ = 33%), indicating relative consistency between studies. No individual study showed a statistically significant result, while the majority of estimates included the unit in the CI.

Furthermore, with regard to SSIs, a comparison was made between the INFL and the CNTRL groups, based on 20 studies (Appendix A). On analysis, the overall RR was 0.95 (95% CI: 0.68 to 1.32), with no statistically significant difference (*p* = 0.74), suggesting no clear effect of the intervention. Heterogeneity was high (I^2^ = 81%), indicating significant variability between studies and possibly different population or intervention characteristics. No individual study showed a statistically significant result, with most showing wide confidence intervals.

Finally, data from 14 studies comparing the likelihood of anastomotic leakage between the INFL and the CNTRL group were analyzed. More precisely, the overall RR was 0.86 (95% CI: 0.50 to 1.48), with no statistically significant difference (*p* = 0.59), suggesting no clear beneficial or detrimental effect of the intervention. Heterogeneity was moderate (I^2^ = 44%), indicating moderate variation between studies. Although some studies showed a trend toward increased or decreased risk, the confidence intervals were wide and most estimates were not statistically significant (Appendix A).

The individual comparative analyses for the subgroups between INFL and CNTRL are attached in Appendix A. More precisely, the administration of INFL was associated with a lower risk of leak in patients with Crohn’s disease. In all other sub-analyses, no significant difference was detected between the comparison groups, as appeared at Appendix A.

#### 3.1.2. VDLZ vs. Controls

The VDLZ treatment group was compared with a CNTRL group in seven studies [34,35,37,43,44,45,47], including a total of 1527 patients (330 VDLZ group and 1197 CNTRL group). The overall effect showed a trend toward VDLZ (RR = 1.26, 95% CI: 0.94–1.67), but the difference was not statistically significant (*p* = 0.12) (Figure 3). Heterogeneity was moderate (I^2^ = 68%, *p* = 0.005), indicating some disagreement between the studies. More precisely, studies such as Kotze et al. [37] (RR = 2.02, 95% CI: 1.16–3.54) and Lightner 2019 [45] (RR = 3.19, 95% CI: 1.12–9.13) showed a statistical difference in favor of VDLZ, in contrast to studies such as Yamada et al. [43], which reported neutral or opposite results. Therefore, overall, the data do not allow for a safe conclusion regarding the clear superiority of VDLZ over the CNTRL group.

In order to further analyze the above (sub-analysis), the forest plot which appears in Appendix A compares the effect of VDZL versus the CNTRL group, with the primary outcome being postoperative ileus in six studies (410 participants in the VDLZ group and 1020 in the control group), with a total of 84 and 208 events, respectively. The overall risk ratio (RR) was 1.41 (95% CI: 0.92–2.17, *p* = 0.11), suggesting only a trend in favor of the VDLZ group. Heterogeneity between studies was moderate (I^2^ = 57%, *p* = 0.04), indicating a unsignificant variation in results. Some studies [34] and [38] showed statistically significant results in favor of the intervention, while others [43] showed the opposite or insignificant differences. Overall, the findings suggest a possible clinical benefit of the intervention, but the lack of statistical significance and the presence of heterogeneity limit the certainty of the conclusions.

Subsequently, the probability of infectious complications (SSIs) occurring between the two groups (VDLZ and CNTRL) was analyzed in seven studies with a total of 444 participants in the experimental group and 1217 in the control group, recording 93 and 300 events, respectively. The RR was 1.28 (95% CI: 0.72–2.28, *p* = 0.40), showing a trend in favor of the experimental intervention, but without a statistically significant difference. Heterogeneity was high (I^2^ = 76%, *p* < 0.001), indicating significant variation between studies. Although some studies [38,40,44] showed results in favor of VDLZ, others [43] and [35] reported results in favor of the CNTRL group, or no significant differences. Overall, there was no strong evidence in favor of the effectiveness of the intervention, mainly due to high heterogeneity and the absence of statistical significance in the overall result.

Finally, the sub-analysis of the five studies compared the incidence of anastomotic leakage between the VDLZ (385 individuals) and the CNTRL group (1027 individuals) (S3.A2). A total of only 8 and 34 events were recorded, respectively, indicating a very low incidence of anastomotic leakage complication. The overall RR was calculated at 0.68 (95% CI: 0.29–1.59, *p* = 0.37), with no statistically significant difference between the groups. Heterogeneity was zero (I^2^ = 0%), reinforcing the consistency of the results across studies. Although some studies showed a trend in favor of the VDLZ group [38,44], the width of the confidence intervals and the small number of events limit the possibility of drawing firm conclusions. Therefore, no significant benefit or risk from the intervention on this outcome is documented.

#### 3.1.3. USTK vs. Controls

The meta-analysis of the two studies [44,47] showed no statistically significant differences between USTK administration and the Control group (RR = 0.55, 95% CI; 0.20–1.57, *p* = 0.26) (Figure 4). Despite the significant difference in Shah’s study [44], the high heterogeneity (I^2^ = 78%) and the non-statistically significant overall effect indicate the need for further research.

As for the risk of postoperative ileus between the two groups, the overall estimated risk ratio (RR) was 0.69 (95% CI: 0.20–2.39), suggesting a non-statistically significant reduction in risk in the USTK group (*p* = 0.56) (Appendix A). The heterogeneity between studies was moderate (I^2^ = 61%), but without a statistically significant difference (*p* = 0.11). The results show that USTK intervention was not associated with a clear reduction or increase in risk compared to the Control group, as the confidence interval crosses the unit and the overall result were not statistically significant.

On the other hand, the comparison concerning the incidence of Surgical Site Infections (SSIs) demonstrated that USTK administration was associated with a significantly reduced risk of that kind of infection compared to the Control group. More precisely, the overall risk ratio was 0.35 (95% CI: 0.17–0.73), indicating a statistically significant reduction in risk in the USTK group (*p* = 0.005). The heterogeneity between studies was zero (I^2^ = 0%), which supported the homogeneity of the results. (Appendix A).

Finally, no clear conclusion could be drawn regarding the effect of USTK intervention on anastomotic leakage rates, due to both the high heterogeneity and uncertainty of the estimated effect, as shown in Appendix A. The overall risk ratio was 6.43 (95% CI: 0.04–958.68), with a particularly wide confidence interval and a statistically insignificant result (*p* = 0.47). The heterogeneity between studies was high (I^2^ = 85%, *p* = 0.01), indicating significant variation in the results of individual studies.

### 3.2. Head-to-Head Analysis in Between the Biological Groups

#### 3.2.1. USTK vs. INFL

The effect of USTK in relation to INFL was compared in a meta-analysis of four studies [44,45,46,47], including in total 721 patients; 114 USTK and 607 INFL. Although there was a trend in favor of USTK (RR = 0.64, 95%CI: 0.31–1.35), the result was not statistically significant (*p* = 0.24). The heterogeneity between the studies was moderate (I^2^ = 64%, *p* = 0.04), indicating that differences between them may affect the validity of the overall conclusion. It is noteworthy that the study by Shah et al. [44] showed statistically significantly better results with USTK (RR = 0.27, 95% CI: 0.13–0.59), which may reflect differences in the population or intervention conditions. Therefore, further comparable studies are needed to assess the potential complications between the two groups (Figure 5).

The forest plot in the sub-analysis comparing the incidence of postoperative ileus rates between the groups of USTK and INFL in three studies (Appendix A) resulted in no clear advantage for either intervention between the groups, as the overall risk ratio was 0.86 (95% CI: 0.47–1.56, *p* = 0.62). Heterogeneity was zero (I^2^ = 0%, *p* = 0.41), indicating homogeneity of results between studies.

Moreover, with regard to surgical infections, the data do not show a statistically significant superiority of USTK over INFL (Appendix A). The overall risk ratio was 0.43 (95% CI: 0.16–1.20), indicating a trend in favor of the USTK group, but without a statistically significant difference (*p* = 0.11). The heterogeneity between studies was moderate (I^2^ = 44%, *p* = 0.17), indicating some variation, but not significant.

Finally, comparing the impact of USTK and INFL administration on anastomotic leakage, the overall results of the three studies (Appendix A) did not support a clear superiority of either group. More precisely, the RR was 1.24 (95% CI: 0.25–6.21), with no statistically significant difference between the two groups (*p* = 0.79). The confidence interval was wide and included the unit, indicating significant uncertainty regarding the direction of the result. Heterogeneity was low (I^2^ = 8%, *p* = 0.34), indicating homogeneity between the studies.

#### 3.2.2. USTK vs. VDLZ

The meta-analysis of four studies [44,45,47,49] compared the effectiveness of perioperative administration of USTK versus VDLZ, in 309 patients (124 and 185, respectively). Overall, USTK was associated with a lower risk of postoperative adverse events (RR = 0.57, 95% CI: 0.26–1.27), but without a statistically significant difference (*p* = 0.17). The heterogeneity of the studies at I^2^ = 62% indicated differences in the results. It is worth nothing that the study of Shah et al. [44] showed statistically better results (USTK: RR = 0.25, 95% CI: 0.11–0.53), while El Aziz [47] showed no significant advantage (RR = 2.19, 95% CI: 0.54–8.91). These conflicting results thus demonstrate the need for further studies (Figure 6).

As shown in the forest plot of the sub-analysis (Appendix A), which compares the incidence of postoperative ileus between the USTK and VDLZ groups in two studies, the overall RR was 0.80 (95% CI: 0.08–8.06), with no statistically significant difference between the groups (*p* = 0.85). The confidence interval was extremely wide and includes the unit, indicating great uncertainty surrounding the result. Heterogeneity was high (I^2^ = 72%, *p* = 0.06), suggesting significant differences between the individual studies. Therefore, no clear conclusion can be drawn regarding the superiority of one intervention over the other as for the incidence of ileus.

Subsequently, regarding the difference in surgical infections between the USTK and VDLZ groups, no clear and reliable conclusion could be drawn about the superiority of one intervention over the other due to the high heterogeneity and uncertainty of the estimates. More precisely, the overall risk ratio was 1.02 (95% CI: 0.10–10.87), with no statistically significant difference between the groups (*p* = 0.99). The very wide confidence interval indicated considerable uncertainty in the estimation of the effect. Heterogeneity was very high (I^2^ = 93%, *p* < 0.00001), indicating marked differences between studies (Appendix A).

#### 3.2.3. INFL vs. VDLZ

Seven studies were analyzed in order to compare the efficacy of INFL versus VDLZ [33,34,36,38,39,40,42,43]. The studies involved 1058 patients, 441 with INFL and 617 VDLZ. The overall result showed no statistically significant difference between the two treatments (RR = 0.92, 95%CI: 0.56–1.52, *p* = 0.76) (Figure 7). However, very high heterogeneity was observed, I^2^ = 89% (*p* < 0.00001), indicating significant differences between studies in terms of outcomes. Some studies [38,39] have demonstrated the superiority of VDLZ (RR = 2.53; 95% CI: 1.60–4.02), whereas others have reported greater efficacy with INFL (RR = 0.36; 95% CI: 0.27–0.49) [40]. This inconsistency reinforces the need for further research.

A meta-analysis of five studies compares the risk of postoperative ileus between INFL and VDLZ treatments (Appendix A). With a total of 315 patients in the INFL group and 487 in the VDLZ group, the overall relative risk (RR) was 2.29 (95% CI: 1.59–3.29, *p* < 0.00001), indicating a significantly higher risk of ileus with VDLZ, compared to INFL. The absence of heterogeneity (I^2^ = 0%) reinforces the reliability of the results. Therefore, treatment with VDLZ appears to be associated with a significantly higher risk of postoperative ileus compared to INFL, which is clinically important for treatment selection in patients undergoing surgery.

We then made a meta-analysis of seven studies and compared the risk of surgical site infections between INFL and VDLZ treatments (Appendix A). With a total of 441 patients in the INFL group and 683 in the VDLZ group, the overall relative risk (RR) was 1.07 (95% CI: 0.65–1.76, *p* = 0.79), with no statistically significant difference between the two groups. Moderate heterogeneity (I^2^ = 47%) suggests moderate variation in study results. Overall, the data did not support a significant difference in the risk of surgical site infections between INFL and VDLZ therapies, suggesting that both therapies have a similar safety profile with regard to this adverse event.

Finally, a comparison was made between the two groups in terms of the incidence of anastomotic leakage (Appendix A). The total sample included 324 patients in the INFL group and 452 in the VDLZ group, with only 3 and 8 cases, respectively. The RR was 0.80 (95% CI: 0.22–2.92, *p* = 0.74), indicating no statistically significant difference in risk between the two treatments. Heterogeneity was zero (I^2^ = 0%), indicating homogeneity in the results, but the very small number of events limited the ability to draw firm conclusions. Therefore, there was no documented difference in the likelihood of anastomotic leakage between INFL and VDLZ.

## 4. Discussion

This current systematic review provides a comparative evaluation of the perioperative safety associated with three commonly used biologic agents for IBD, namely the Infliximab, the Vedolizumab and the Ustekinumab. As the application of biologics has expanded in the treatment of IBDs, a considerable number of patients are now undergoing surgery while on treatment. This evolving situation has raised important concerns regarding the potential impact of biologic agents on surgical outcomes.

Our findings suggest that USTK shows the most favorable perioperative safety profile among the three medications, revealing no significant increase in overall or specific postoperative complications in the studies analyzed. However, the limited number of studies examining the USTK factor should be acknowledged, as it may affect the reliability of the results. Conversely, notwithstanding the variability in findings, VDLZ showed, in some studies, a trend towards elevated rates of postoperative ileus and inflammatory complications, such as mucosal fistulas and anastomotic leakage, particularly when administered close to the surgery date. While INFL has been debated concerning its safety in surgical settings, this review uncovered a mainly neutral profile, with most studies indicating no meaningful rise in postoperative risk relative to control groups.

These results highlight important differences in the surgical risk profiles of biologic therapies and emphasize the need for individualized perioperative planning, taking into account factors such as timing of administration, disease severity, and concurrent immunosuppressive therapies.

### 4.1. Inflammatory Bowel Diseases and Biological Therapies

IBDs are idiopathic diseases caused by immune dysregulation, where the body responds to the host’s intestinal microbiota. Their diagnosis rate has tended to increase over the years. The two main types of IBDs are UC, which is confined to the colon, and CD, which can involve any part of the GI tract. These diseases have a clear genetic predisposition, and patients are more prone to developing a variety of malignancies [50]. In several studies, genetic factors appear to influence the risk of IBDs by causing impaired integrity of the epithelial barrier [51], deficiencies in autophagy and pattern recognition receptors, and problems in lymphocyte differentiation, particularly in Crohn’s disease [52].

A variety of inflammatory mediators have been identified in IBD patients and have been shown to play a key role in the pathological and clinical features of these disorders. More precisely, cytokines, which are released by macrophages in response to various antigenic stimuli, bind to different receptors and promote autocrine, paracrine and endocrine responses. In addition, cytokines differentiate lymphocytes into different types of T cells. Helper T cells, type 1 (Th-1), are mainly associated with Crohn’s disease, whereas type 2 (Th-2) cells are mainly associated with ulcerative colitis. The immune response generated therefore disrupts the intestinal mucosa and leads to acute–neutrophilic- and chronic–lymphocytic, histiocytic–inflammatory process [53].

Moreover, the association between the environment and the development of inflammatory bowel disease is not clear. A significant and more ‘stable’ role is attributable to smoking, which has a negative effect on the development of Crohn’s disease, in contrast to Ulcerative Colitis. Furthermore, eating habits are thought to be partly involved in the development of IBDs. In particular, it appears that a diet rich in fiber and the consumption of fruits and vegetables has a protective effect on the intestinal mucosa, whereas high meat consumption has a negative effect and is associated with a potentially higher incidence of inflammatory bowel disease [54].

Although, in the pre-biologic era up to 80% of patients required surgery to manage the disease and its complications [55], since the introduction of the anti-TNF biological agents, the need for surgical intervention has significantly decreased [56]. Subsequently, early initiation of an anti-TNF in combination with an immunomodulator factor has significantly reduced the rates of surgery, hospitalization, or serious disease-related complications in the 2 years following the induction of treatment [57]. However, one-third of patients with IBDs do not respond to anti-TNF agents due to non-TNF-mediated inflammatory pathways, and even one-third of initially responsive patients lose the ability to respond to therapy due to both reduced drug levels or the development of antibodies against them [58]. Until 2014, non-anti-TNF therapies for these diseases were limited, until the approval of VDLZ, an α4β7 integrin inhibitor. In 2016, USTK, a monoclonal antibody targeting Interleukins 12 and 23, was approved for the treatment of moderate to severe Crohn’s disease [59]; therefore, today approximately 30–50% of patients are likely to be receiving a biologic agent at the time of surgery [60].

### 4.2. Infliximab

INFL is a chimeric monoclonal antibody that inhibits tumor necrosis factor-alpha (TNF-α), a cytokine involved in systemic inflammation. It is indicated for the treatment of moderate to severe Crohn’s disease, and ulcerative colitis. By binding to TNF-α, INFL prevents it from interacting with its receptors, thereby reducing inflammation and altering disease progression. INFL is administered intravenously, typically in a healthcare setting. Initial dosing may be followed by maintenance doses at specified intervals [61]

Of the articles collected, 30 studied the effect of INFL in the perioperative period. The articles involved a total of 16,731 patients, of whom 701 had IBD (8 studies), 10,604 patients had ulcerative colitis (UC) (10 studies), and 5426 patients had Crohn’s disease (12 studies). The average time between the last dose of the drug and surgery was 6–8 weeks for 1945 patients in 5 studies, 12 weeks for 4946 patients in 9 studies, and 4 weeks for 477 patients in 3 studies.

Overall, in the majority of studies, no difference was observed in the rate of postoperative complications between patients treated with INFL and those with no preoperative treatment or an alternative formulation. More precisely, 20 of the 30 studies concluded that treatment with the biological agent of INFL did not relate to surgical complications, or that any differences were statistically insignificant [13,14,16,17,18,19,20,23,24,25,27,28,29,34,37,38,39,40,41,42,43]. In one study, an increase in the number of days of hospitalization was observed, but with no difference in overall complications [30]. Furthermore, in another study with 151 patients, a difference in the frequency of postoperative complication rate was observed only in cases of co-administration of INFL with cyclosporine [31]. Finally, the study concluded that although there were no differences in overall complication rate between groups, the patients who received INFL treatment preoperatively exhibited an increase in surgical wound infections [21]. It should be noted that in this study, the administration of INFL preceded the surgical intervention by approximately 25 weeks; however, the authors report that the time interval between these two is in no case related to INFL use.

In contrast, in 5 of the total studies on the postoperative effect of INFL, an increase in surgical complications was observed. More specifically, a study in patients with UC, who were divided into two groups depending on whether they underwent one-stage or two-stage surgery [22], concluded that patients who received INFL followed by one-stage surgery had increased rates of pelvic inflammation and sepsis, in contrast to patients who underwent two-stage surgery, in whom there was no overall difference in postoperative complications from the group of patients without preoperative treatment. Additionally, pouch related complications were also reported after INFL administration to 47 patients with UC, either as an independent factor, or, when co-administrated with other immunosuppressants (azathioprine) or corticosteroids [32]. Moreover, low albumin levels and malnutrition affect the adverse effects of anti-TNF administration [33,36,44], as does corticosteroid use [33].

Finally, the administration of INFL reduced the rates of surgical site infections in patients with UC compared to the CNTRL group [26], but without clear statistical significance (*p* = 0.181), while overall postoperative complications remained at the same levels in both groups. Uchino et al. [18,19] also reported that patients in three INFL group exhibited fewer SSIs.

### 4.3. Vedolizumab

VDLZ is a humanized monoclonal antibody targeting the α4β7 integrin, which is integral to the migration of leukocytes into the GI tract. It is approved for use in both Crohn’s disease and ulcerative colitis. By blocking the interaction between α4β7 integrin and mucosal addressing cell adhesion molecule-1 (MAdCAM-1), VDLZ effectively inhibits the gut-selective homing of inflammatory cells, thereby reducing local inflammation without systemic immune suppression. It is administered intravenously, similar to INFL, initially requiring a loading dose followed by maintenance infusions [41].

With regard to the agent of VDLZ, 12 articles were included, involving a total of 3551 patients. One study referred to patients with Crohn’s disease (312 patients) [38], three studies involved 496 patients diagnosed with ulcerative colitis [34,40,42], and finally, the remaining eight studies investigated the effects of VDLZ postoperatively in a total of 2743 patients with IBD [33,35,36,38,41,43,62,63]. The time interval between the administration of VDLZ and surgery was 4 weeks for 3 studies [33,36,43], 12 weeks for 7 of the 12 articles [34,35,38,39,40,41,63], while Ferrante [42] examined the effects of the biological agent 16 weeks after administration. It should be noted that in 6 studies, the comparison of postoperative complications after the administration of biological agents was made between three subgroups of patients (anti-TNF agent/INFL, VDLZ, and CNTRL group) [34,35,39,41,42,43]. Four studies compared postoperative complications between VDLZ and anti-TNF agent/INFL [33,36,38,40], one study compared patients receiving VDLZ exclusively with the CNTRL group [63], and finally, in one of the 12 studies, the occurrence of complications was assessed in relation to the circulating levels of VDLZ at the time of surgery [62].

As for the primary conclusions of the studies, in four of them no difference was observed in the occurrence of postoperative complications [33,35,36,63], although the study by Kotze PG et al. [63] concerned surgeries outside the GI tract. The majority of studies observed an increase in postoperative complications in patients receiving VDLZ, particularly in terms of inflammatory complications or postoperative ileus. To be more precise, Parrish et al., Kim et al., and Lightner et al. [34,39,62] observed an increase in the rate of postoperative ileus in the VDLZ subgroup. Subsequently, four studies reported a significant increase in complications related to the surgical field (*p* = 0.002) [38,39] as well as the development of mucosal fistulas after stoma diversion [38,39,40,41], but also with increased rates of anastomotic leakage [40]. In contrast, in two of the twelve studies, a positive effect was observed after preoperative administration of VDLZ: Ferrante et al. [42] in 2017 observed a reduction in overall complications in this group of patients 16 weeks after administration of the agent, while in the same year, Yamada et al. [43] reported a reduction in inflammatory complications (3.1%) in VDLZ patients, as well as a reduction in more general complications related to surgical intervention.

Consequently, we find considerable heterogeneity and inconsistency between studies, without being able to draw any clear conclusions about the impact of Vedolizumab administration, particularly with regard to the likelihood of ileus occurring.

### 4.4. Ustekinumab

Finally, USTK is an interleukin-12/23 (IL-12/23) inhibitor that has received approval for the treatment of moderate to severe Crohn’s disease as well as plaque psoriasis. By binding to the p40 subunit shared by IL-12 and IL-23, USTK disrupts the signaling pathways that contribute to inflammation and immune response [64].

A total of 5 studies referring to complications after USTK administration [44,45,46,47,49] were collected, all involving patients with Crohn’s disease (144 in total). In 2 of the 5 studies, patients were divided into 4 groups, receiving anti-TNF, VDLZ, USTK, and a Control group, respectively [44,47]. The other three studies compared patients receiving USTK or INFL [45,46], or VDLZ [45,49], respectively. In all studies, the respective biologic agent was administered up to 12 weeks prior to surgery. None of the studies reported a statistically significant difference in the incidence of postoperative complications, with the unique exception of the study by Shah RS [44], which identified an increased incidence of postoperative intra-abdominal abscesses in VDLZ-treated patients. Crohn’s disease is considered as an independent risk factor for opportunistic surgical site infections, while smoking or elevated glucose levels increase the risk of infections related to intra-abdominal organs (organ space SSIs). In addition, cortisone, with last dose given 4 weeks prior to surgery, as well as obesity, increase the risk of infections, while the co-administration of immunosuppressants also increases the risk of sepsis [3].

### 4.5. Summary and Clinical Implications

The present Systematic Review provides a comparative overview of postoperative surgical complications associated with the three commonly used biologic agents—INFL, VDLZ, and USTK—in IBD patients undergoing intestinal surgery. The findings suggest that ΙΝFL and USTK are generally not associated with a significant increase in postoperative complications, while VDLZ shows a trend towards a higher incidence of inflammatory complications and postoperative ileus, particularly when administered shortly before surgery. However, given the variability of the studies, these results cannot be considered sufficient to draw definitive conclusions.

Although the safety of INFL in the perioperative period has been debated, most included studies found no significant association with increased surgical morbidity, aligning with previous meta-analyses and large cohort studies. USTK appears to have the most favorable perioperative safety profile, as none of the included studies showed a statistically significant increase in complications—a finding that may be attributed to its more targeted mechanism and lower systemic immunosuppression; nevertheless, this number remains too limited to draw clear, general conclusions. In contrast, VDLZ, despite its gut-selective mechanism, was paradoxically associated in several studies with delayed healing and increased local complications. This may raise the question of whether gut-selectivity does not necessarily equate to improved perioperative outcomes, possibly due to its role in mucosal immune regulation and impaired leukocyte trafficking during tissue repair. However, it remains merely a hypothesis, given the heterogeneity of the studies.

Finally, across all treatment groups, corticosteroid use within 4 weeks of surgery, malnutrition, obesity, and combination immunosuppressive therapy were consistently reported as independent risk factors for adverse outcomes. These findings emphasize the importance of individualized perioperative risk assessment, taking into account both the biologic agent and the patient’s overall clinical status.

### 4.6. Limitations

Several limitations must be acknowledged. First, the included studies were largely retrospective cohorts, introducing selection bias and limiting causal inference. Second, heterogeneity in patient populations, disease severity, type of surgery, and definitions of postoperative complications poses challenges for uniform interpretation. Third, timing of drug discontinuation prior to surgery varied significantly across studies, which may have impacted outcomes. Additionally, circulating drug levels and therapeutic drug monitoring were not uniformly reported, limiting our understanding of the dose–response relationship. We should also point out once again that the studies concerning the group undergoing treatment with USTK were very few in number, a fact that further weakens our conclusions. Finally, it should be noted that the three biological factors are not synchronous. During the 15 years of their use, anesthesia and surgical techniques and tools, as well as antibiotics, have improved. In addition, most of the surgical procedures are now performed by means of laparoscopy. All the above should be considered as factors affecting the complication rates.

## 5. Conclusions

This systematic review supports the hypothesis that when IBD patients, already undergoing INFL and USTK treatment, are subjected to intestine surgery, the complication rate was similar to the general population. However, there seems to be a relation between VDLZ and more frequently occurring inflammatory complications and postoperative ileus, especially when given shortly before surgery. Further prospective, multicenter studies are warranted to better delineate the perioperative risk profiles of biologic agents, ideally stratifying patients by disease phenotype, nutritional status, and concomitant immunosuppression. The role of therapeutic drug monitoring in preoperative planning remains an area of interest, especially for agents like VDLZ, where drug levels may predict outcomes. A standardized approach to the timing of biologic discontinuation relative to surgery would be greatly beneficial in establishing clinical guidelines.

## Figures and Tables

**Figure 1 pharmaceuticals-18-01466-f001:**
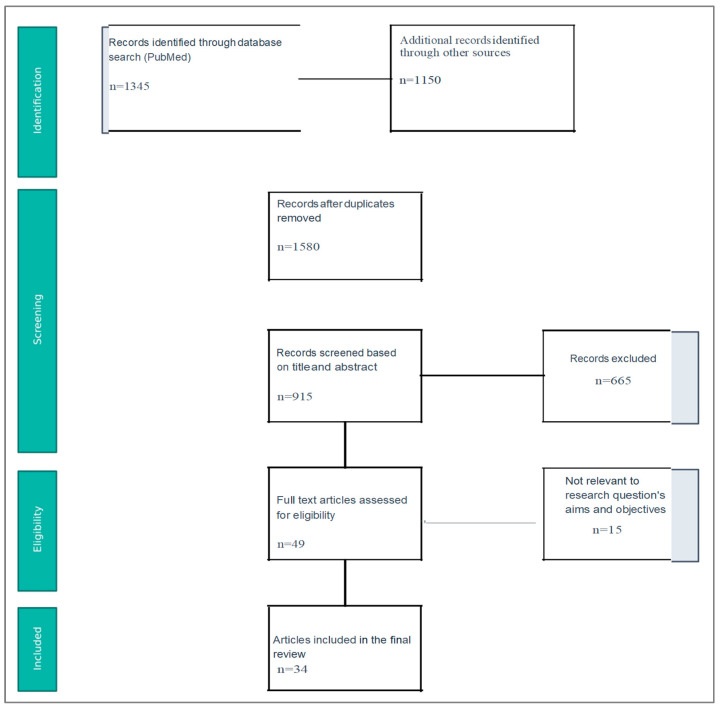
Flow Chart.

**Figure 2 pharmaceuticals-18-01466-f002:**
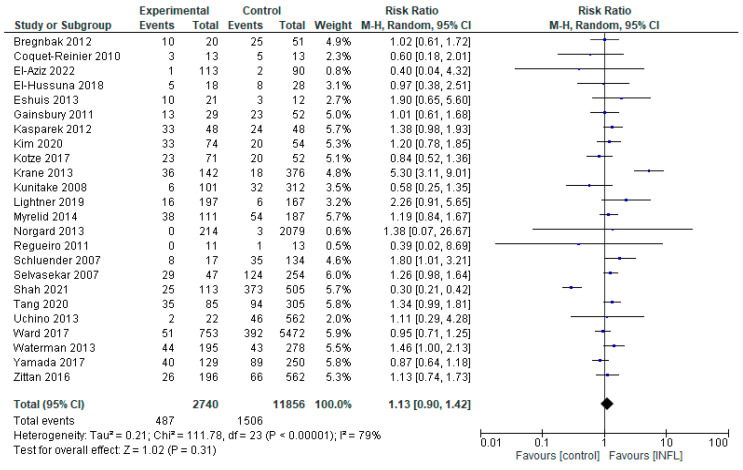
Total complications (studies [13,14,15,16,17,18,19,20,21,22,23,24,25,26,27,28,29,30,31,32,34,37,43,44,45,47]); INFL vs. CNTRL.

**Figure 3 pharmaceuticals-18-01466-f003:**
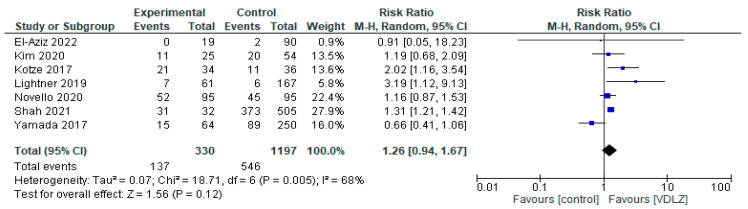
Total complications (studies [34,35,37,43,44,45,47]); VDLZ vs. CNTRL.

**Figure 4 pharmaceuticals-18-01466-f004:**
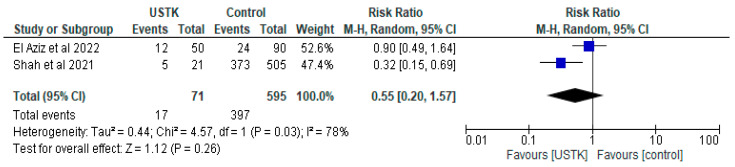
Total complications (studies [44,47]); USTK vs. Controls.

**Figure 5 pharmaceuticals-18-01466-f005:**
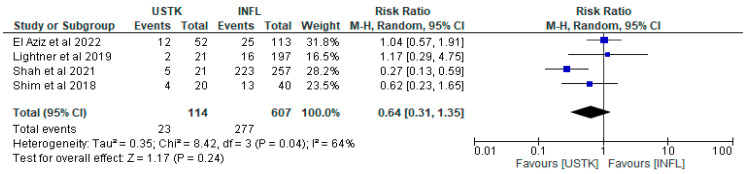
Total complications (studies [44,45,46,47]); USTK vs. INFL.

**Figure 6 pharmaceuticals-18-01466-f006:**
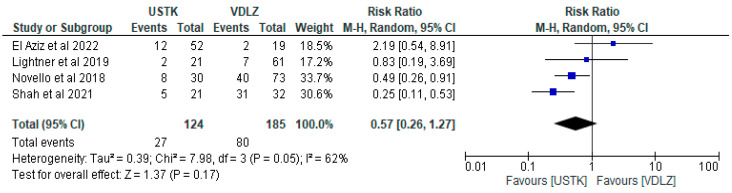
Total complications (studies [44,45,47,49]); USTK vs. VDLZ.

**Figure 7 pharmaceuticals-18-01466-f007:**
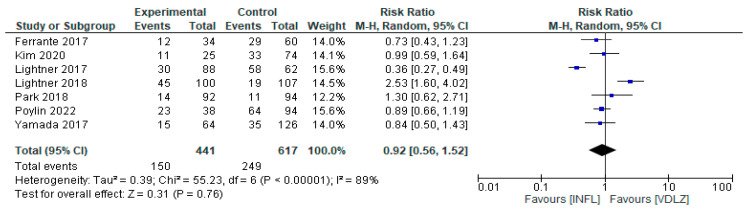
Total complications (studies [33,34,36,38,39,40,42,43]); INFL vs. VDLZ.

**Table 1 pharmaceuticals-18-01466-t001:** Baseline Characteristics of Included Studies.

Study		Study Design	Type of IBD	Patients (n)	Time from Last Biologic Infusion to Surgery, Days [Median (IQR)]
1	Tang, Shasha et al. (2020) [13]	Retrospective	CD	390	8 weeks
2	El-Hussuna, Alaa et al. (2018) [14]	Prospective Multi-Center Pilot Study	CD/UC	46	12 weeks
3	Ward, S. T et al. (2017) [15]	Population Study	UC	753	12 weeks (4 weeks_418 pts)
4	Zittan, Eran et al. (2016) [16]	Retrospective	UC	773	<15 or 15–30 or 30–180 days
5	Myrelid, P. et al. (2014) [17]	Multicenter Study	CD	298	2 months
6	Uchino, Motoi et al. (2013) [18]	Prospective	CD	405	43 days
7	Uchino, Motoi et al. (2013) [19]	Clinical Trial	UC	196	12 weeks
8	Krane, Mukta K et al. (2013) [20]	Retrospective	IBDs	142	
9	Waterman, Matti et al. (2013) [21]	Retrospective	IBDs	195	<14 or 15–30 or 31–180 days
10	Eshuis, Emma J et al. (2013) [22]	Retrospective	UC	72	7 months
11	Norgard, B M et al. (2013) [23]	Cohort	CD	2293	12 weeks
12	Kasparek, M. S et al. (2012) [24]	Prospective	CD	48	3 months
13	Norgard, B M et al. (2012) [25]	Cohort	UC	1226	12 weeks
14	Bregnbak, D. et al. (2012) [26]	Retrospective	UC	71	90 days
15	Regueiro, Miguel et al. (2011) [27]	Double Blinded Controlled Study	CD	24	4 weeks
16	Gainsbury, M. L et al. (2011) [28]	Retrospective	UC	81	12 weeks
17	Coquet-Reinier, B. et al. (2010) [29]	Retrospective	UC	26	44 days
18	Kunitake, Hiroko et al. (2008) [30]	Retrospective	CD/UC	413	12 weeks
19	Schluender, Stefanie J et al. (2007) [31]	Prospective	UC	151	2 months
20	Selvasekar, C. R et al. (2007) [32]	Retrospective	UC	301	2 months
21	Poylin, Vitaliy Y et al. (2022) [33]	Retrospective	CD/UC	199	4 weeks
22	Kim, Jeong Yeon et al. (2020) [34]	Matched Case–Control Study	UC	153	12 weeks
23	Novello, M et al. (2020) [35]	Matched Case–Control Study	CD/UC	980	12 weeks
24	Park, K T et al. (2018) [36]	Retrospective Cohort	CD/UC	186	30 days
25	Kotze, P.G. et al. (2017) [37]	Case-Matched Study	CD/UC	68	12 weeks
26	Lightner, Amy L et al. (2018) [38]	Retrospective Cohort	UC	435	12 weeks
27	Lightner, A L et al. (2018) [39]	Retrospective	CD	100	12 weeks
28	Lightner, Amy L et al. (2017) [40,41]	Retrospective	UC	150	12 weeks
29	Ferrante, Marc et al. (2017) [42]	Retrospective	UC	170	8–16 weeks
30	Yamada, Akihiro et al. (2017) [43]	Retrospective	CD/UC	443	4 weeks
31	Shah, Ravi S et al. (2021) [44]	Retrospective	CD	815	12 weeks
32	Lightner, Amy L et al. (2019) [45]	Retrospective	CD	712	12 weeks
33	Shim, Hang Hocj et al. (2018) [46]	Cohort	CD	60	4 months
34	Abd El Aziz, Mohamed A et al. (2022) [47]	Retrospective	CD	274	4 weeks

## Data Availability

Data presented in this study is contained within the article and Appendix A. Further inquiries can be directed to the corresponding author.

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
