# Peer review of "Comparative Risk of Complications Following Intestinal Surgery After Infliximab, Vedolizumab, or Ustekinumab Treatment: Systematic Review & Meta-Analysis"

_pharmaceuticals, 2025, doi:10.3390/ph18101466_

Round 1
Reviewer 1 Report
Comments and Suggestions for Authors
Dear Authors,
There are a few issues that need to be addressed before this paper is accepted for publcication. For example:
The introduction is too lengthy and partially repetitive; please streamline while ensuring all key prior systematic reviews are cited.
Expand your search strategy (include EMBASE, Scopus, Cochrane). Provide a PRISMA checklist and register the review.
Clarify inclusion/exclusion criteria: how were mixed UC/CD studies handled? Provide a funnel plot/publication bias analysis.
Avoid overstating conclusions, especially regarding Ustekinumab (data too sparse) and Vedolizumab (results inconsistent).
Improve figures: at least one or two forest plots should appear in the main text. Simplify Table 1 (move excessive detail to supplement). Furthemore, you need to substantially revise the English for both clarity and grammar.
Lastly, ensure consistency in terminology (e.g., “USTK,” “INFL,” “VDLZ” vs full names). and consider acknowledging the limitations more frankly: retrospective nature of studies, very low number of Ustekinumab cases, heterogeneity.
Author Response
We would like to thank the reviewer for the valuable comments. We have uploaded a file containing our point-by-point responses

Reviewer 2 Report
Comments and Suggestions for Authors
A Network Meta-Analysis was conducted to compare the risk of complications following intestinal surgery after treatment with Infliximab, Vedolizumab, or Ustekinumab. The study concluded that Ustekinumab demonstrated the most favorable safety profile, while Vedolizumab was associated with higher rates of ileus and inflammatory complications. These conclusions are well-supported by the results. Given the increasing use of biologics in the treatment of IBD, this research offers valuable insights that can benefit clinical practice. I recommend publishing this study in its current form.
Author Response
We would like to thank the reviewer for the valuable comments on our manuscript and we appreciate the time given for reading our work.
Reviewer 3 Report
Comments and Suggestions for Authors
This manuscript presents a network meta-analysis comparing perioperative safety profiles of Infliximab, Vedolizumab, and Ustekinumab in patients undergoing intestinal surgery for inflammatory bowel disease. The topic is timely and clinically relevant, given the expanding use of biologics in IBD patients requiring surgical intervention. The study adheres to PRISMA guidelines and synthesizes evidence from 34 studies, offering indirect comparative insights in the absence of head-to-head randomized trials. The study provides comprehensive literature search and inclusion of a large number of studies with critical findinsg suggesting that Ustekinumab may offer a favorable safety profile, while Vedolizumab could be associated with increased postoperative ileus. There are some significant concerns related to the study inclusion and interpretation as summarized below:
1- Most of the included studies are retrospective in nature, introducing selection bias and limiting causal inference.
2-The analysis is further challenged by high heterogeneity across studies, with I² values often exceeding 70%, which undermines the reliability of pooled estimates.
3-Event rates for certain outcomes such as anastomotic leakage and fistula formation are very low, producing wide confidence intervals and unstable risk estimates. Additionally, variability in the timing of biologic discontinuation prior to surgery (ranging from 4 to 16 weeks) complicates interpretation and may confound outcomes, particularly in subgroup analyses. Historical changes in surgical practice—such as the shift toward laparoscopy, improved anesthetic management, and advances in perioperative care over the 15-year span of included studies—may also bias the results.
4-Finally, the absence of consistent reporting on therapeutic drug monitoring or circulating drug levels limits mechanistic understanding, and several sections of the manuscript suffer from grammatical errors, awkward phrasing, and inconsistent tense, which diminish clarity and readability.
Please include these limitations in a limitations section
Author Response
We are grateful to the reviewer for the constructive feedback. Please find our detailed, point-by-point responses in the uploaded document

Round 2
Reviewer 1 Report
Comments and Suggestions for Authors
The authors have addressed some of the issues raised in my first review. The manuscript is improved in clarity & structure, but several methodological & statistical concerns remain. These need further attention before the paper can be accepted.
Remaining Major Issues
- Risk of Bias No structured risk of bias assessment is presented. A validated tool (e.g., Newcastle–Ottawa Scale, ROBINS-I) should be applied and results tabulated (main or Supplementary).
- Heterogeneity High I² values persist. Sensitivity and subgroup analyses are mentioned but not sufficiently detailed. Please provide full stratified analyses (Crohn’s vs UC, infusion timing, prospective vs retrospective studies) with forest plots in the Supplementary.
- Publication Bias Funnel plots and Egger’s tests are not presented. These should be provided for outcomes with ≥10 studies.
- Outcome Definitions Greater transparency is needed on how heterogeneous definitions (e.g., infectious complications, ileus) were harmonized across studies. If definitions diverged, specify whether such studies were excluded.
- Network Meta-Analysis Claim The title and text describe a “network meta-analysis,” but the manuscript still presents largely pairwise pooled results. If a true NMA was performed, league tables and SUCRA rankings should be included. If not, the manuscript should be retitled and reframed as a comparative meta-analysis.
Minor Issues
- Abstract should note that most included studies were retrospective.
- Table 1 requires standardized reporting (Crohn’s vs UC, infusion timing).
- The discussion of vedolizumab and ileus remains speculative; emphasize inconsistency.
- Some sentences in the Introduction/Discussion remain overly long and should be simplified.
Author Response
We appreciate all the reviewers' valuable comments. We have revised the manuscript as suggested. You will find attached the point-by-point answer to each comment.

Reviewer 3 Report
Comments and Suggestions for Authors
Accept in present form
Author Response
We would like to thank the reviewer for the valuable comments, that have improved the quality of our manuscript.
Round 3
Reviewer 1 Report
Comments and Suggestions for Authors
The authors have clearly improved the manuscript in both structure and methodology. They streamlined the introduction, expanded the database search and registered the review (with PRISMA checklist), and added funnel plots as requested. A structured risk of bias assessment (QUIPS) is now presented, with results tabulated in the supplement. Subgroup and sensitivity analyses (Crohn’s vs UC, timing, design) have been provided with supporting forest plots. Outcome definitions are more clearly harmonized, and the mislabeling of “network meta-analysis” has been corrected to “comparative meta-analysis.” Figures and tables are clearer, with forest plots included in the main text and excessive detail shifted to the supplement. Language has been revised, and the limitations section more frankly acknowledges heterogeneity, retrospective study designs, and the small number of ustekinumab studies.
Overall, the revision addresses the major concerns I raised. Some heterogeneity and interpretive limitations inevitably remain, but these are now explicitly discussed. The manuscript is substantially stronger and, in my view, suitable for acceptance after these changes.